# Epidemiology and Molecular Characterization of *Entamoeba* spp. in Non-Human Primates in Zoos in China

**DOI:** 10.3390/vetsci11120590

**Published:** 2024-11-25

**Authors:** Diya An, Shui Yu, Tingting Jiang, Jianhui Zhang, Qun Liu, Jing Liu

**Affiliations:** 1National Animal Protozoa Laboratory, Key Laboratory of Animal Epidemiology of the Ministry of Agriculture and Rural Affairs, College of Veterinary Medicine, China Agricultural University, Beijing 100193, China; sy20203050847@cau.edu.cn (D.A.); yushui@cau.edu.cn (S.Y.); sy20223051011@cau.edu.cn (T.J.); qunliu@cau.edu.cn (Q.L.); 2Northeast Agricultural University Library, Harbin 150030, China; zhangjianhui@neau.edu.cn; 3National Key Laboratory of Veterinary Public Health and Safety, College of Veterinary Medicine, China Agricultural University, Beijing 100193, China

**Keywords:** *Entamoeba* spp., non-human primates, epidemiology, genotype, zoonoses

## Abstract

Captive animals in zoos, particularly non-human primates (NHPs), are in close contact with humans, raising concerns about the transmission of zoonotic diseases. NHPs, being phylogenetically similar to humans, are susceptible to infections by various species of *Entamoeba* (*Entamoeba* spp.). The aim of this study was to investigate the prevalence of *Entamoeba* spp. in captive NHPs in Chinese zoos. We collected fecal samples from 14 NHP species in five regions of China and examined them for six *Entamoeba* species. The results show that three asymptomatic *Entamoeba* species capable of infecting humans, *Entamoeba coli*, *E. dispar*, and *E. polecki*, were prevalent among NHPs. This indicates a potential zoonotic risk and underscores the need to strengthen control measures for asymptomatic parasites in zoos to prevent cross-infection between humans and animals.

## 1. Introduction

*Entamoeba* spp. are present in humans, NHPs, and various other vertebrate and invertebrate species around the world [1]. At least seven species have been identified as parasitizing the human gut, including *E. coli*, *E. dispar*, *E. histolytica*, *E. hartmanni*, *E. moshkovskii*, *E. polecki*, and *E. Bangladeshi* [2,3]. The global molecular prevalence of *Entamoeba* spp. infections in humans is 3.55% (3817/107,396), and amebiasis, caused by *E. histolytica*, is the second most common parasitic disease-related cause of death worldwide, resulting in about 67,900 deaths per year [4,5,6]. Although the age-standardized disability-adjusted life years rate of *Entamoeba* spp. infection-associated diseases presented significantly declining trends, it has remained a heavy burden among the age group of <5 years and the low sociodemographic index regions from 1990 to 2019 [7], raising concerns about potential zoonotic transmission.

Due to space constraints and close contact with humans, the lifestyle of zoo animals differs significantly from that of their wild counterparts. In captivity, animals are more frequently exposed to feces and fecal-contaminated food and water, which increases the risk of disease in captive animals and poses a potential threat to animal caretakers and visitors. *Entamoeba* spp. are frequently reported as protozoa parasites in captive NHPs [8,9], and there are six main species of intestinal *Entamoeba* spp. to which NHPs are susceptible, including *E. chattoni*, *E. coli*, *E. dispar*, *E. hartmanni*, *E. nuttalli*, and *E. polecki* [6]. Few molecular epidemiological studies of NHPs have been published, with most conducted in Asia, Europe, Africa, and North America [4]. It has been shown that NHPs can be experimentally infected with *E. histolytica* cysts of human origin without developing invasive disease [7]. Some studies suggest that lemurs in both the wild and in zoo settings may be infected with *E. histolytica*, resulting in diarrhea symptoms [8,9]. The relationship between the pathogenicity of *Entamoeba* spp. in NHPs and zoonotic diseases still needs to be further explored.

The morphological similarity among *Entamoeba* spp. in the intestine makes it challenging to differentiate them using microscopy alone, particularly between *E. histolytica*, *E. moshkovskii*, and *E. dispar* [10]. Therefore, various molecular techniques, including PCR, nested PCR, real-time PCR, multiplex PCR, and loop-mediated isothermal amplification (LAMP), have been widely used in epidemiologic investigations because of their high sensitivity and specificity [11,12]. The small subunit ribosomal RNA (SSU rRNA) gene is a multicopy gene that is relatively easy to amplify from fecal samples, providing sufficient resolution to distinguish between *Entamoeba* spp. Evidence for the genetic diversity of *Entamoeba* spp. in NHPs is primarily based on SSU rRNA gene analyses, differential diagnosis by PCR, and characterization of the SSU rRNA gene [13].

In this study, we used PCR to amplify SSU rRNA gene loci from six *Entamoeba* species, including *E. coli*, *E. dispar*, *E. histolytica*, *E. moshkovskii*, *E. nuttalli*, and *E. polecki*, to explore *Entamoeba* spp. and their zoonotic potential in 14 species of NHPs.

## 2. Materials and Methods

### 2.1. Specimen Collection

Samples were collected on the basis of whether the animals could come into direct or indirect contact with people, taking into account factors such as visitor feeding, fecal disposal, and enclosures. A total of 84 fecal samples were randomly collected from NHPs in Beijing (n = 34), Guiyang (n = 10), Shijiazhuang (n = 18), Tangshan (n = 9), and Xingtai (n = 13) in China from September 2020 to November 2021. The NHP species included in this study were *Ateles fusciceps* (brown-headed spider monkeys, n = 1), *Colobus polykomos* (king colobus, n = 6), *Erythrocebus patas* (patas monkey, n = 3), *Lemur catta* (ring-tailed lemur, n = 21), *Macaca leonina* (pig-tailed macaques, n = 2), *Macaca mulatta* (rhesus macaque, n = 10), *Mandrillus sphinx* (mandrills, n = 9), *Nomascus annamensis* (northern yellow-cheeked crested gibbon, n = 10), *Pan troglodytes* (chimpanzees, n = 9), *Papio hamadryas* (baboon, n = 2), *Rhinopithecus roxellana* (golden snub-nosed monkeys, n = 5), *Saimiri sciureus* (squirrel monkeys, n = 3), *Sapajus apella* (capuchin monkeys, n = 1), and *Trachypithecus francoisi* (Francois’ langur, n = 2), totaling 14 species of NHPs (Table 1). All fecal samples were stored independently at −20 °C for subsequent testing.

### 2.2. Genomic DNA Extraction

TIANamp Stool DNA Kit (Tigen, Beijing, China) was used for fecal samples genomic DNA extraction. According to the instructions, 200 mg of fecal sample was used for genomic DNA extraction, and finally, DNA was eluted with 50 mL elution buffer. The extracted DNA samples were stored at −20 °C for reserve.

### 2.3. PCR Amplification

Eighty-four collected fecal DNA samples from NHPs were tested by PCR for *Entamoeba* spp. (*E. nuttalli* [14], *E. coli* [14], *E. polecki* [14], *E. dispar* [15], *E. histolytica* [15], and *E. moshkovskii* [15]). The amplification targets were SSU rRNA gene locus sequences. The upstream primer of *E. dispar*, *E. histolytica*, and *E. moshkovskii* was Enta F, while downstream primers differed. The primer sequences are shown in Table 2.

### 2.4. Sequencing and Phylogenetic Analysis

Positive PCR products were sequenced (Ruiboxingke Company, Beijing, China) and compared with published/reference sequences in GenBank to determine the sample species/genotype using the BLAST tool (https://blast.ncbi.nlm.nih.gov/Blast.cgi) and Clustal X 2.13 software.

The reference sequences downloaded from GenBank and the SSU rRNA gene locus sequences obtained in this study were used to construct an evolutionary tree based on the neighbor-joining (NJ) method, and the Tamura–Nei model using Mega11.0.13 software and bootstrap analysis with 1000 replicates were performed.

### 2.5. Statistical Analysis

Results were processed into contingency tables according to factors such as location, species, and detection status. Since the proportion of cells with an expected count of fewer than 5 is >20%, we chose Fisher’s exact test for statistical analysis using SPSS 26.0 software, as well as the 95% confidence intervals (CIs) of detection rates. A statistical significance level of *p* < 0.05 was considered to indicate a significant difference.

## 3. Results

### 3.1. Occurrence of Entamoeba spp.

In this study, we used six species-specific primers to detect *Entamoeba* spp. in the fecal samples of 14 NHP species from five local zoos. A total of 19 out of 84 fecal samples tested positive, for an overall positive rate of 22.6% (95% CI: 15.0–32.7%). Only 3 species of *Entamoeba* spp. were detected: *E. coli*, *E. dispar*, and *E. polecki*. Among them, *E. coli* had the highest positivity rate of 14.3% (12/84, 95% CI: 8.4–23.3%), followed by *E. dispar* at 8.3% (7/84, 95% CI: 4.1–16.2%), and *E. polecki* at 7.1% (6/84, 95% CI: 3.3–14.7%), *E. histolytica*, *E. moshkovskii*, and *E. nuttalli* were not detected. Additionally, co-detection results indicate that 1 sample (1.2%, 95% CI: 0.2–6.4%) was concurrently detected by three *Entamoeba* spp., 2 samples (2.4%, 95% CI: 0.7–8.3%) and 2 samples (2.4%, 95% CI: 0.7–8.3%) were simultaneously detected by *E. coli* + *E. polecki* and *E. coli* + *E. dispar*, while no simples for *E. polecki* + *E. dispar* were detected (Table 3).

### 3.2. Geographic Distribution of Entamoeba spp.

Table 4 summarizes the prevalence of *Entamoeba* spp. identified among NHPs across the 5 study locations. Positive samples were detected only in fecal samples from Beijing, Guiyang, and Shijiazhuang zoos. The highest positivity rate was observed in Shijiazhuang Zoo at 33.3% (6/18), followed by Beijing Zoo at 29.4% (10/34) and Guiyang Zoo at 30% (3/10). The prevalence of *Entamoeba* spp. varied significantly among zoos (*p* < 0.01). In the Beijing Zoo, three *Entamoeba* species were identified, with *E. coli* showing the highest positivity rate at 26.5% (9/34), followed by *E. dispar* at 11.8% (4/34), and *E. polecki* at 5.9% (2/34). There were also cases of mixed detections: one case involving *E. coli* and *E. polecki*, two cases involving *E. coli* and *E. dispar*, and one case involving all three *Entamoeba* spp. In Shijiazhuang Zoo, *E. coli* and *E. polecki* were detected with positivity rates of 16.7% (3/18) and 22.2% (4/18), respectively, including one case of mixed detection with both species. In Guiyang Zoo, only *E. dispar* detections were found, with a positivity rate of 30% (3/10).

### 3.3. Distribution Patterns of Detections Among Species and Molecular Characterization

Prevalence of *Entamoeba* spp. identified among NHPs according to host species is summarized in Table 5. Out of the 14 NHPs, seven were detected by *Entamoeba* spp., including *Colobus polykomos*, *Erythrocebus patas*, *Lemur catta*, *Mandrillus sphinx*, *Nomascus annamensis*, *Pan troglodytes*, and *Papio hamadryas*. The highest prevalence was observed in *Colobus polykomos* at 83.3% (5/6). *Colobus polykomos* also had the highest rate of *E. coli* detection at 66.7 (4/6). The highest rate of *E. dispar* detection was found in *Erythrocebus patas* at 66.7 (2/3), while *Mandrillus sphinx* had the highest rate of *E. polecki* detection at 44.4 (4/9). The prevalence of *Entamoeba* spp. detections varied significantly among different NHP species (*p* < 0.01). Prevalence of *Entamoeba* spp. identified among NHPs according to zoos and host species is summarized in Table 6; Fisher’s exact test results show that the prevalence of *Entamoeba* spp. in Beijing Zoo and Shijiazhuang Zoo was significantly different among NHPs (*p* < 0.01) but not in Guizhou Zoo NHPs (*p* > 0.01). The constructed phenetic tree (Figure 1) illustrates the assignment of *Entamoeba* spp. within hosts, demonstrating that the affinities of *Entamoeba* spp. were similar among the same species of NHPs. *Mandrillus sphinx* had the highest rates of mixed detections, including one case of *E. coli* and *E. polecki* (*M. sphinx* 68, Beijing), one case of *E. coli* and *E. dispar* (*M. sphinx* 72, Beijing), and one case of mixed detection by all three *Entamoeba* spp. (*M. sphinx* 73, Beijing). In addition, a mixed detection of *E. coli* and *E. dispar* was found in *Papio hamadryas* (*P. hamadryas* 60, Beijing), and a mixed detection of *E. coli* and *E. polecki* was found in *Pan troglodytes* (*P. troglodytes* 33, Shijiazhuang).

## 4. Discussion

China is relatively rich in primate resources and is currently a major producer and primary supplier of NHPs in the international market [16]. *Entamoeba* spp. are among the most common intestinal parasites in NHPs [17], capable of spreading rapidly and causing widespread infections because of their direct monoxenous life cycle, a short prepatent period, and various transmissible morphological forms [18,19,20,21,22]. Despite this, there have been relatively few studies on the infection rates and species distribution of *Entamoeba* spp. in the intestines of NHPs in China.

In this study, SSU rRNA gene locus sequences of six *Entamoeba* spp. include *E. coli*, *E. dispar*, *E. histolytica*, *E. moshkovskii*, *E. nuttalli*, and *E. polecki*. Among these, *E. coli, E. dispar*, and *E. polecki* are known to infect both humans and NHPs [23,24,25,26]. Our findings revealed these three species were present among the NHPs studied, with *E. coli* having the highest prevalence (26.5%) among the five zoos, followed by *E. dispar* (11.8%). These results are consistent with those reported by Dos Santos Zanetti et al., who detected *Entamoeba* spp. in human and animal samples from Brazil [27]. In addition, we observed instances of co-detections, including cases where *E. coli* was found alongside two other *Entamoeba* spp. Such mixed detections, particularly involving *E. coli* and *E. dispar*, are commonly reported in global studies [14,28,29,30,31,32]. While *E. histolytica* and *E. moshkovskii* are more frequently detected in humans, these two species were not identified in our study. *Entamoeba histolytica* is pathogenic and can cause amoebiasis in humans [2,8], but its occurrence in NHPs is rare, with reports limited to a few countries, including China [33,34], Belgium [32], the Netherlands [35], Singapore [36], and the Philippines [37,38]. Recent evidence suggests that what has been previously identified as *E. histolytica* in NHPs is usually a distinct species, *E. nuttalli* [13]. *Entamoeba nuttalli* appears to be prevalent among NHPs and is often associated with sympatric carriage [39]. Feng et al. [30] and Yu et al. [40] reported the presence of *E. nuttalli* in NHPs from the Guangxi, Guiyang, and Sichuan regions of southwest China; however, our experimental results do not detect this species, possibly due to its host-specific distribution to NHPs [41] or regional variability.

The infection rate of wild NHPs *Entamoeba* spp. seems to be higher than that of captive ones, e.g., Wild Macaca mulatta (89.96%, Taihang Mountain area, China) [42] and Pan troglodytes (79%, savanna woodland, Tanzania) [14] had significantly higher positive detection rates of *Entamoeba* spp. than captive NHPs, such as in the Zoological Garden in Belgium (44%) [8] and in Nanjing, China (49.17%) [43]. Our *Entamoeba* spp. detection rate (22.6%) was lower than that of the above studies but higher than that of European zoo NHPs (8.8%) [44] and Ibadan in Nigeria [45]. This may be due to the fact that captive breeding and management in zoos hinder the spread of Entamoeba spp. However, rapid urbanization in recent years has led to the construction of zoos with concrete enclosures or floors, potentially facilitating the accumulation of feces in the animals’ living environments. This may increase the risk of cross-contamination of parasites among groups of animals through the fecal–oral route of transmission. In addition, *Entamoeba* spp. were detected in rats that were either free-living sympatric [46] or used as food for captive animals [47] in zoos, contaminated water, food, contact with shared keepers, or the introduction of infected new animals could further exacerbate this risk [21], these factors raise concerns regarding the health of captive animals in zoos and the risk of zoonotic diseases. Our previous studies have demonstrated that animals in the zoos of Beijing, Guiyang, Shijiazhuang, Tangshan, and Xingtai are affected by a variety of intestinal protozoan infections, including *Cryptosporidium* spp., *Giardia duodenalis*, *Enterocytozoon bieneusi*, and *Blastocystis* spp. [48]. In this study, *Entamoeba* spp. were detected in NHPs from Beijing, Guiyang, and Shijiazhuang zoos. The results show significant differences in the prevalence of *Entamoeba* spp. in different zoos (Table 1); this may be related to factors such as regional prevalence, lifestyle of NHPs, and zoo management. Unfortunately, our study did not capture the relevant information. *Entamoeba* spp. prevalence differs significantly among NHPs (Table 5) and NHPs in Beijing Zoo and Shijiazhuang Zoo (Table 6), with mixed detections observed in Beijing (four cases) and Shijiazhuang (one case) zoos. Phylogenetic analyses showed that phenetic relationships of *Entamoeba* spp. were similar within the same NHPs (Figure 1); this suggests the potential for cross-infection of NHPs in the same environment. In zoos, symptomatic animals typically attract the attention of caretakers, whereas all three *Entamoeba* spp. identified in this study were asymptomatic and detectable in the human gut. This indicates that asymptomatic *Entamoeba* spp. are likely prevalent in NHPs and may be zoonotic, underscoring the need for molecular detection methods and preventive measures to reduce the risk of zoonotic diseases.

Due to the social nature of most NHPs studied, collecting individual fecal samples posed challenges. Consequently, our study lacked analyses correlating *Entamoeba* spp. detections with variables such as sex, age, and symptoms. Among the 14 NHPs investigated, *Entamoeba* spp. were detected in only 7, primarily those whose natural habitats are in Africa and Southeast Asia. *Colobus polykomos* exhibited the highest detection rate (83.3%), predominantly with *E. coli* (4/6, 66.7%) and *E. polecki* (1/6, 16.7%). This aligns with the findings of Roland Yao Wa Kouassi et al. in Taï National Park, Côte d’Ivoire [49], suggesting the importance of Colobus polykomos in preventing detections of *Entamoeba* spp. *Mandrillus sphinx* exhibited the highest rates of mixed detections (3/9, 33.3%); mixed detections appear to be common in *Mandrillus sphinx* and have been recorded in both semi-free-range [50] and wild [51] environments. *Pan troglodytes,* one of the closest evolutionary relatives of humans [52], can be infected with *E. histolytica* and have zoonotic potential [53]. Our experimental results show that *Pan troglodytes* can be detected with *E. coil* (3/9, 33.3%), *E. dispar* (1/9, 11.1%), and *E. polecki* (1/9, 11.1%), with a total detection rate of 44.4% (4/9), with one case of mixed detection of *E. coil* and *E. polecki*. These three *Entamoeba* species seem to be frequently detected in *Pan troglodytes* [54,55,56]. Our study further confirms that mixed *Entamoeba* spp. detections occur in captive *Mandrillus sphinx*, and the transmission dynamics in this species warrant further investigation.

## 5. Conclusions

This study investigated the prevalence of *Entamoeba* spp. in 14 specials of NHPs across five zoos in China. Our findings indicate that the asymptomatic presence of three *Entamoeba* spp. of *E. coli*, *E. dispar*, and *E. polecki* was significantly prevalent among NHPs in those zoos, with a potential risk for cross-contamination. This raises concerns about the increased risk of zoonotic transmission to both humans and other animals. It is crucial to recognize and address asymptomatic parasitic infections in herd animals within zoo environments and implement effective measures to prevent the spread of these parasites.

## Figures and Tables

**Figure 1 vetsci-11-00590-f001:**
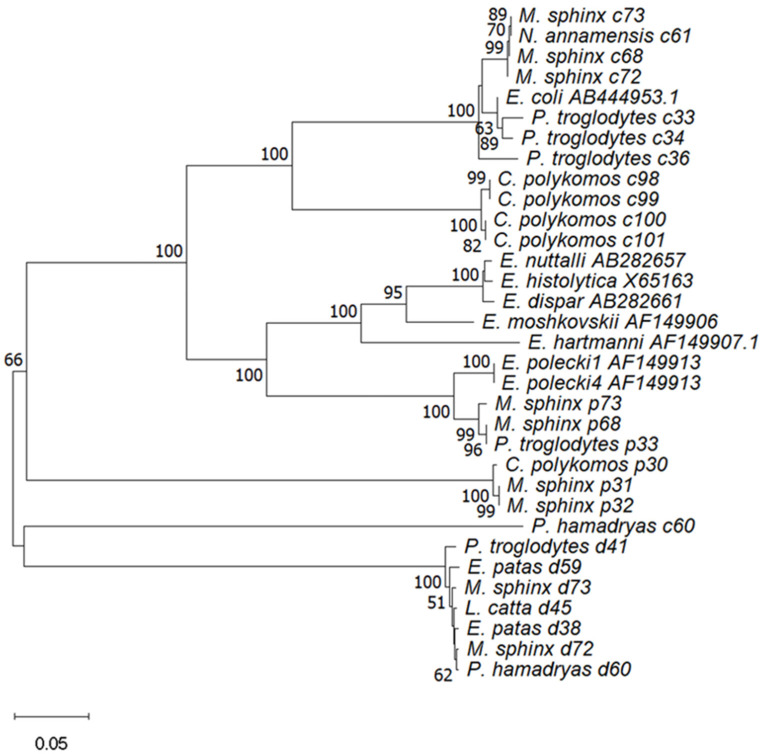
Phenetic relationships of *Entamoeba* spp. Numbers on the branches are percent bootstrapping values from 1000 replicates, only bootstrap values > 50 are indicated. The accession numbers utilized for the identification of *Entamoeba* spp. were AB444953 (*E. coli*), AB282661 (*E. dispar*), AF149907 (*E. hartmanni*), X65163 (*E. histolytica*), AF149906 (*E. moshkoskii*), AB282657 (*E. nuttalli*), AF149913 (*E. polecki-like* variant 1), and AF149912 (*E. polecki-like* variant 4).

**Table 1 vetsci-11-00590-t001:** The distribution of NHPs across 5 studied zoos.

Study Locations	Species	No. Examined
Beijing Zoo	*Ateles fusciceps*	1
	*Colobus polykomos*	4
	*Erythrocebus patas*	1
	*Lemur catta*	10
	*Mandrillus sphinx*	6
	*Nomascus annamensis*	7
	*Papio hamadryas*	1
	*Rhinopithecus roxellana*	3
	*Trachypithecus francoisi*	1
Guiyang Zoo	*Erythrocebus patas*	1
	*Lemur catta*	2
	*Mandrillus sphinx*	1
	*Nomascus annamensis*	1
	*Pan troglodytes*	2
	*Papio hamadryas*	1
	*Rhinopithecus roxellana*	1
	*Trachypithecus francoisi*	1
Shijiazhuang Zoo	*Colobus polykomos*	1
	*Lemur catta*	4
	*Mandrillus sphinx*	2
	*Nomascus annamensis*	2
	*Pan troglodytes*	4
	*Rhinopithecus roxellana*	1
	*Saimiri sciureus*	3
	*Sapajus apella*	1
Tangshan Zoo	*Colobus polykomos*	1
	*Erythrocebus patas*	1
	*Lemur catta*	2
	*Macaca leonina*	2
	*Pan troglodytes*	3
Xingtai Zoo	*Lemur catta*	3
	*Macaca mulatta*	10
Total		84

**Table 2 vetsci-11-00590-t002:** Species-specific primers used in diagnostic PCR for *Entamoeba* spp.

Primers	Specificity	Product Size (bp)
E. nF: ATTTTATACATTTTGAAGACTTTGCAE. nR: CTCTAACCGAAATTAGATAACTAC	*E. nuttalli* [14]	840
E. cF: GAAGCTGCGAACGGCTCATTACE. cR: CACCTTGGTAAGCCACTACC	*E. coli* [14]	290
E. pF: GGAAGGCTCATTATAACAGTTATAGE. pR: CCTCATTATTATCCTATGCTTC	*E. polecki* [14]	680
Enta F: ATGCACGAGAGCGAAAGCAT		
E. pR: CACCACTTACTATCCCTACC	*E. dispar* [15]	752
E. hR: GATCTAGAAACAATGCTTCTCT	*E. histolytica* [15]	166
E. mR: TGACCGGAGCCAGAGACAT	*E. moshkovskii* [15]	580

**Table 3 vetsci-11-00590-t003:** Occurrence of *Entamoeba* spp. in NHPs of 5 study zoos.

Entamoeba spp.	No. Positive/Samples	Prevalence (%)	95% CI (%)	Scientific Name (No. Positive)
*E. coli*	12/84	14.3	8.4–23.3	*Colobus polykomos* (4), *Mandrillus sphinx* (3), *Nomascus annamensis* (1), *Pan troglodytes* (3), *Papio hamadryas* (1)
*E. dispar*	7/84	8.3	4.1–16.2	*Erythrocebus patas* (2), *Lemur catta* (1), *Mandrillus sphinx* (2), *Pan troglodytes* (1), *Papio hamadryas* (1)
*E. polecki*	6/84	7.1	3.3–14.7	*Colobus polykomos* (1), *Mandrillus sphinx* (4), *Pan troglodytes* (1)
*E. coli* + *E. polecki* (only)	2/84	2.4	0.7–8.3	*Mandrillus sphinx* (1), *Pan troglodytes* (1)
*E. coli* + *E. dispar* (only)	2/84	2.4	0.7–8.3	*Mandrillus sphinx* (1), *Papio hamadryas* (1)
*E. coli + E. dispar* + *E. polecki*	1/84	1.2	0.2–6.4	*Mandrillus sphinx* (1)
Total	19/84	22.6	15.0–32.7	

**Table 4 vetsci-11-00590-t004:** Prevalence of *Entamoeba* spp. identified among NHPs across 5 study zoos.

Study Locations	No. Examined	No. Positive (%)	*f*	*p* Value *	Type of *Entamoeba* spp. Identified		
					Number of Positive Samples (%)		
					*E. coli*	*E. dispar*	*E. polecki*
Beijing Zoo	34	10 (29.4)	9.236	0.041 *	9 (26.5)	4 (11.8)	2 (5.9)
Guiyang Zoo	10	3 (30.0)			0	3 (30.0)	0
Shijiazhuang Zoo	18	6 (33.3)			3 (16.7)	0	4 (22.2)
Tangshan Zoo	9	0			0	0	0
Xingtai Zoo	13	0			0	0	0
Total	84	19 (22.6)			12 (14.3)	7 (8.3)	6 (7.1)

*f*: Fisher’s exact test. *: Significant at 0.05.

**Table 5 vetsci-11-00590-t005:** Prevalence of *Entamoeba* spp. identified among NHPs according to host species.

Host Species	No. Examined	No. Positive (%)	*f*	*p* Value *	Type of *Entamoeba* spp. Identified		
					Number of Positive Samples (%)		
					*E. coli*	*E. dispar*	*E. polecki*
*Ateles fusciceps*	1	0	31.549	<0.05 *	0	0	0
*Colobus polykomos*	6	5 (83.3)			4 (66.7)	0	1 (16.7)
*Erythrocebus patas*	3	2 (66.7)			0	2 (66.7)	0
*Lemur catta*	21	1 (4.8)			0	1 (4.8)	0
*Macaca leonina*	2	0			0	0	0
*Macaca mulatta*	10	0			0	0	0
*Mandrillus sphinx*	9	5 (55.6)			3 (33.3)	2 (22.2)	4 (44.4)
*Nomascus annamensis*	10	1 (10.0)			1 (10.0)	0	0
*Pan troglodytes*	9	4 (44.4)			3 (33.3)	1 (11.1)	1 (11.1)
*Papio hamadryas*	2	1 (50.0)			1 (50.0)	1 (50.0)	0
*Rhinopithecus roxellana*	5	0			0	0	0
*Saimiri sciureus*	3	0			0	0	0
*Sapajus apella*	1	0			0	0	0
*Trachypithecus francoisi*	2	0			0	0	0
Total	84	19 (22.6)			12 (14.3)	7 (8.3)	6 (7.1)

*f*: Fisher’s exact test. *: Significant at 0.05.

**Table 6 vetsci-11-00590-t006:** Prevalence of *Entamoeba* spp. identified among NHPs according to zoos and host species.

Study Locations	Species	No. Examined	No. Positive (%)	*f*	*p* Value *	Type of *Entamoeba* spp. Identified		
						Number of Positive Samples (%)		
						*E. coli*	*E. dispar*	*E. polecki*
Beijing Zoo	*Ateles fusciceps*	1	0	20.375	<0.05 *	0	0	0
	*Colobus polykomos*	4	4 (100.0)			4 (100.0)	0	0
	*Erythrocebus patas*	1	1 (100.0)			0	1 (100.0)	0
	*Lemur catta*	10	0			0	0	0
	*Mandrillus sphinx*	6	3 (50.0)			3 (50.0)	2 (33.3)	2 (33.3)
	*Nomascus annamensis*	7	1 (14.3)			c	0	0
	*Papio hamadryas*	1	1 (100.0)			1 (100.0)	1 (100.0)	0
	*Rhinopithecus roxellana*	3	0			0	0	0
	*Trachypithecus francoisi*	1	0			0	0	0
Guiyang Zoo	*Erythrocebus patas*	1	1 (100.0)	5.778	1.000	0	1 (100.0)	0
	*Lemur catta*	2	1 (50.0)			0	1 (50.0)	0
	*Mandrillus sphinx*	1	0			0	0	0
	*Nomascus annamensis*	1	0			0	0	0
	*Pan troglodytes*	2	1 (50.0)			0	1 (50.0)	0
	*Papio hamadryas*	1	0			0	0	0
	*Rhinopithecus roxellana*	1	0			0	0	0
	*Trachypithecus francoisi*	1	0			0	0	0
Shijiazhuang Zoo	*Colobus polykomos*	1	1 (100.0)	12.182	0.017 *	0	0	1 (100.0)
	*Lemur catta*	4	0			0	0	0
	*Mandrillus sphinx*	2	2 (100.0)			0	0	2 (100.0)
	*Nomascus annamensis*	2	0			0	0	0
	*Pan troglodytes*	4	3 (75.0)			3 (75.0)	0	1 (25.0)
	*Rhinopithecus roxellana*	1	0			0	0	0
	*Saimiri sciureus*	3	0			0	0	0
	*Sapajus apella*	1	0			0	0	0
Total		84	19 (22.6)			12 (14.3)	7 (8.3)	6 (7.1)

*f*: Fisher’s exact test. *: Significant at 0.05.

## Data Availability

The sequences that support the findings of this study are openly available in the GenBank database at https://www.ncbi.nlm.nih.gov/nucleotide/ (accessed on 3 June 2024).

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
