# Peer review of "Epidemiology and Molecular Characterization of Entamoeba spp. in Non-Human Primates in Zoos in China"

_vetsci, 2024, doi:10.3390/vetsci11120590_

Round 1
Reviewer 1 Report
Comments and Suggestions for Authors
The manuscript “Epidemiology and Molecular Characterization of Entamoeba spp Infection in Non-human Primates of Zoos in China” is relatively simple and may not be very appealing to an international audience, though it addresses a relevant topic in the local context.
However, several points should be considered to improve the article.
-In addition to the scientific names, the common names of the animals should be included when available, to allow for broader understanding.
-The distribution of species across each zoo should be presented, along with the inclusion criteria of zoo and animals in the study.
-Were all non-primate species from the zoos included? If not, the reasons for their exclusion should be clarified. Including images of the studied animals would also be desirable.
-The use of the chi-square test is mentioned, but it is unclear what was being compared. All comparisons should be explicitly detailed, and the theoretical basis for these comparisons should be justified in the introduction.
-Given the small sample size, Fisher’s exact test may be more appropriate, and the authors should explore this option and present the corresponding results.
-The presentation of the statistical results should be improved. Detailed explanations of all comparisons should be provided, as there is currently only a general mention of significance. This needs to be broken down, and p-values should be included in the tables.
-The results of statistical comparisons also need to be discussed in more depth.
-The discussion is quite generic, with limited reference to the results, and it does not address the limitations of the study, which need to be discussed.
Comments on the Quality of English Language
The writing quality requires attention, as there are issues with semantics and grammar that should be corrected.
Author Response
Dear Editors and Reviewers,
We are grateful for all the valuable comments and suggestions from the reviewers, all of which have been addressed by adding additional experiments, explanations and discussions as well as figure removal and rearrangement. Point-by-point responses are below.
Comments 1: In addition to the scientific names, the common names of the animals should be included when available, to allow for broader understanding.
Response 1: We agree with this comment. Therefore, we have added common names of the NHPs to help readers better understand this article. [Line 83-90]
Comments 2: The distribution of species across each zoo should be presented, along with the inclusion criteria of zoo and animals in the study.
Response 2: Thank you for pointing this out. Our study picks up where the previous study left off and addresses the epidemiology of Entamoeba spp. infections in NHPs in these five zoos, the distribution of species across each zoo has been proposed in previous study (Epidemiology and Molecular Characterization of Zoonotic Gastrointestinal Protozoal Infection in Zoo Animals in China. Animals: an open access journal from MDPI 2024, 14, doi:10.3390/ani14060853).
Comments 3: Were all non-primate species from the zoos included? If not, the reasons for their exclusion should be clarified. Including images of the studied animals would also be desirable.
Response 3: Not all NHPs were included, some primates that are not on public display were not collected, and the samples we collected were considered for direct or indirect contact with humans, and the reasons for exclusion were added to the manuscript. [Line 78-80] But unfortunately the zoo did not provide images of the studied animals, we will emphasise this point in subsequent studies, thank you very much for your comments!
Comments 4: The use of the chi-square test is mentioned, but it is unclear what was being compared. All comparisons should be explicitly detailed, and the theoretical basis for these comparisons should be justified in the introduction.
Response 4: We are very sorry for our negligence of statistical analysis, we have added data to Table 4-6 to clarify the comparative information, and added the specific methodology and selection rationale for the statistical analysis to the manuscript. [Line 116-120]
Comments 5: Given the small sample size, Fisher’s exact test may be more appropriate, and the authors should explore this option and present the corresponding results.
Response 5: Thank you very much for pointing out our error, the chi-square test in the manuscript has been replaced with Fisher's exact test and the results have been added to the manuscript. [Table 4-6]
Comments 6: The presentation of the statistical results should be improved. Detailed explanations of all comparisons should be provided, as there is currently only a general mention of significance. This needs to be broken down, and p values should be included in the tables.
Response 6: Thank you for your valuable comments, in order to provide a more detailed explanation, we have added the occurrence of Entamoeba spp. in NHPs of 5 study zoos [Table 3, line 134], broken down the distribution of samples collected from each zoo in NHPs [Table 1, line 92] and presented their infections [Table 6, line 176] in the manuscript, and the p values have also been added to the tables.
Comments 7: The results of statistical comparisons also need to be discussed in more depth.
Response 7: Agree. We discuss the results of the statistical comparisons in more depth and add them to the manuscript. [Line 233-236, 237-240]
Comments 8: The discussion is quite generic, with limited reference to the results, and it does not address the limitations of the study, which need to be discussed.
Response 8: We have re-written this section and added some references [Line 213-220, 250-261] to better illustrate the results, and discussed the limitations of the study. [Line 223-236, 246-248]
Comments 9: The writing quality requires attention, as there are issues with semantics and grammar that should be corrected.
Response 9: We are sorry for the language problem and have corrected it in according to your suggestions. In addition, we have asked native speakers to touch up and revise the entire manuscript to prevent grammatical and semantic problems.
We sincerely appreciate the time and effort invested by the reviewers in evaluating our manuscript. We look forward to any additional feedback or suggestions. Special thanks to you for your good comments.
Yours Sincerely,
Jing Liu
Reviewer 2 Report
Comments and Suggestions for Authors
Dear Authors
Greetings
The epidemiology and molecular characterization of Entamoeba species, particularly in zoo environments, is a fascinating and important area of study. These protozoan parasites are known to infect a variety of hosts, including humans and animals, leading to gastrointestinal illnesses. However I consider that your work can be improved.
1th. The title
2th. Keywords If the authors will talk about epidemiology the keyword EPIDEMIOLOGY must to be listed
3th Use a higher % of recent articles
example: Prevalence and Molecular Identification of Entamoeba spp. in Non-human Primates in a Zoological Garden in Nanjing, China - PMC (nih.gov)
Novel Entamoeba Findings in Nonhuman Primates: Trends in Parasitology (cell.com)
Molecular characterization and zoonotic potential of Entamoeba spp., Enterocytozoon bieneusi and Blastocystis from captive wild animals in northwest China | BMC Veterinary Research | Full Text (biomedcentral.com)
Characterization of Entamoeba fatty acid elongases; validation as targets and provision of promising leads for new drugs against amebiasis | PLOS Pathogens
Molecular Characterization of Entamoeba spp. in Wild Taihangshan Macaques (Macaca mulatta tcheliensis) in China - PubMed (nih.gov)
For instance, a study conducted in a wildlife sanctuary in northwest China identified seven different Entamoeba species, with a high overall infection rate of 54.97% among the sampled animals. Another study in the Nanjing Hongshan Forest Zoo found that non-human primates had a high prevalence of Entamoeba infections, with species like Entamoeba histolytica, E. dispar, and E. coli being commonly detected. These findings underscore the importance of continuous surveillance and molecular characterization to understand the transmission dynamics and potential zoonotic risks associated with Entamoeba species in zoo environments. This knowledge is vital for implementing effective control measures to protect both animal and human health.
Epidemiological investigation of Entamoeba in wild rhesus macaques in China: A novel ribosomal lineage and genetic differentiation of Entamoeba nuttalli - ScienceDirect
Global burden and trends of the Entamoeba infection-associated diseases from 1990 to 2019: An observational trend study - ScienceDirect
Entamoeba histolytica Infection in the Philippines: A Review (mbimph.com)
At the discussion the authors would can insert comparasions to better justify the results
An Annotated Checklist of the Human and Animal Entamoeba (Amoebida: Endamoebidae) Species- A Review Article - PMC (nih.gov)
Entamoeba - an overview | ScienceDirect Topics
Growth and genetic manipulation of Entamoeba histolytica - PMC (nih.gov)
https://www.cambridge.org/core/journals/epidemiology-and-infection/article/entamoeba-histolytica-in-wild-rats-caught-in-london/C6221AF38DF9BF6D592B06BED71EB7FF#:~:text=A%20description%20is%20given%20of%20active%20forms
(maybe the rats can be a important point to the risks in one zoo...)
Other important thing is to use graphical results to make your work inovative in comparasion with another version published or similar studies.
Kind regards
Comments on the Quality of English Language
Minor editing of English language required.
Author Response
Dear Editors and Reviewers,
We are grateful for all the valuable comments and suggestions from the reviewers, all of which have been addressed by adding additional experiments, explanations and discussions as well as figure removal and rearrangement. Point-by-point responses are below.
Comments 1: The title
Response 1: Thank you very much for your suggestion, I wonder if you could provide more detailed instructions to help us change the title, we will complete your suggestion in subsequent revision!
Comments 2: Keywords If the authors will talk about epidemiology the keyword EPIDEMIOLOGY must to be listed
Response 2: We agree with this comment. Therefore, we have added Epidemiology to keywords. [Line 35]
Comments 3: Use a higher % of recent articles
Response 3: We sincerely appreciate the valuable comments. We have checked the literature carefully and added references into the Introduction part and Discussion part in the revised manuscript, in addition, we have added other literature to enrich our article and support the ideas.
The references to each literature are as follows:
- Prevalence and Molecular Identification of Entamoeba spp. in Non-human Primates in a Zoological Garden in Nanjing, China - PMC (nih.gov) [Line 217, reference 43]
- Novel Entamoeba Findings in Nonhuman Primates: Trends in Parasitology (cell.com) [Line 72, reference 13]
- Molecular characterization and zoonotic potential of Entamoeba spp., Enterocytozoon bieneusi and Blastocystis from captive wild animals in northwest China | BMC Veterinary Research | Full Text (biomedcentral.com)
Response: We have carefully read all the literature and developed a new idea of comparing the prevalence of Entamoeba spp. in wild and captive NHPs, which has been added to the manuscript, [Line 213-220] and we have cited similar literature to complement this aspect. Thank you very much for your suggestions.
- Characterization of Entamoeba fatty acid elongases; validation as targets and provision of promising leads for new drugs against amebiasis | PLOS Pathogens
Response: The content of this literature is of low relevance to this manuscript, and we regret that we have not cited it in the revised manuscript.
- Molecular Characterization of Entamoeba spp. in Wild Taihangshan Macaques (Macaca mulatta tcheliensis) in China - PubMed (nih.gov) [Line 214, reference 44]
- Epidemiological investigation of Entamoeba in wild rhesus macaques in China: A novel ribosomal lineage and genetic differentiation of Entamoeba nuttalli – ScienceDirect [Line 209, reference 40]
- Global burden and trends of the Entamoeba infection-associated diseases from 1990 to 2019: An observational trend study - ScienceDirect [Line 47 and 59, reference 7]
- Entamoeba histolytica Infection in the Philippines: A Review (mbimph.com) [Line 206, reference 37]
- An Annotated Checklist of the Human and Animal Entamoeba (Amoebida: Endamoebidae) Species- A Review Article - PMC (nih.gov) [Line 195, reference 26]
- Entamoeba - an overview | ScienceDirect Topics
Response: Unfortunately, we did not retrieve this literature, could we trouble you to provide more detailed information to help us re-retrieve this literature? We will add this article in a subsequent revision.
- Growth and genetic manipulation of Entamoeba histolytica - PMC (nih.gov)
Response: We cite its references “Global, regional, and national life expectancy, all-cause mortality, and cause-specific mortality for 249 causes of death, 1980-2015: a systematic analysis for the Global Burden of Disease Study 2015” to enrich the content of the manuscript. [Line 55, reference 6]
- https://www.cambridge.org/core/journals/epidemiology-and-infection/article/entamoeba-histolytica-in-wild-rats-caught-in-london/C6221AF38DF9BF6D592B06BED71EB7FF#:~:text=A%20description%20is%20given%20of%20active%20forms
Response: We cite other similar literature to add to your point “maybe the rats can be an important point to the risks in one zoo...” [Line 225, reference 46 and 47]
Comments 4: Other important thing is to use graphical results to make your work inovative in comparasion with another version published or similar studies.
Response 4: Thank you very much for your suggestion, however due to the complexity of the data we have not been able to form intuitive graphical results, given this, we have added the occurrence of Entamoeba spp. in NHPs of 5 study zoos [Table 3, line 134], broken down the distribution of samples collected from each zoo in NHPs [Table 1, line 92] and presented their infections [Table 6, line 176] in the manuscript to make the results more intuitive.
We sincerely appreciate the time and effort invested by the reviewers in evaluating our manuscript. We look forward to any additional feedback or suggestions. Special thanks to you for your good comments.
Yours Sincerely,
Jing Liu
Reviewer 3 Report
Comments and Suggestions for Authors
The manuscript is very important to understand Entamoeba spp prevalence in primates in chinese Zoos. I found the manuscript very easy to read, and deeply interesting because of the importance of this infective protozoans i close related species to humans. I think statistical analyses could be improved, and the description of the species by study locations must be detailed. Here I mention some of the mistakes detected and suggestions to the authors.
Line 20: correct throughout the manuscript E. coli, (E. coil is wrong).
Line 26: “in the intestines” not directly.
Line 28: spp.(space)by
Line 41: (.) instead of (,) “[2,3], The global”
Line 110: It could be interesting to do a more complex statistical analysis…you could try a GLM (binomial for prevalence) to figure it out if there is some difference between places or interactions. It is not clear to me if the difference between species is because there is a source of contagion in one place and not because a single species is more prone to get a protozoan. To improve the results section (or even in methods) it could helpful to add a table with species/individuals/study location.
It is also a pity you didn’t count oocysts in fecal samples by Macmaster devices or similar tool as a complement, to calculate intensities that could be helpful to understand the process.
Line 178: should start with the full name if it is a dot in the final. “, E. nuttalli [11]. Entamoeba nuttalli
Line 210: “(p < 0.01)” this is out of place, it’s a result.
In Discussion or Conclusion: don’t you think is important to discuss the genetic proximity of Pan troglodytes to Homo sapiens, considering the risks of contagion of close related protozoans?
Author Response
Dear Editors and Reviewers,
We are grateful for all the valuable comments and suggestions from the reviewers, all of which have been addressed by adding additional experiments, explanations and discussions as well as figure removal and rearrangement. Point-by-point responses are below.
Comments 1: Line 20: correct throughout the manuscript E. coli, (E. coil is wrong).
Response 1: We sincerely thank you for careful reading, we have corrected and checked the entire manuscript to make sure that similar errors do not occur! [Line 20, 29, 126, 144, 167]
Comments 2: Line 26: “in the intestines” not directly.
Response 2: Agree. We have, accordingly, changed “in the intestines” to “in the fecal samples” to make the description more accurate. [Line 26]
Comments 3: Line 28: spp.(space)by
Response 3: Thank you very much for pointing out our error, which we have corrected in the manuscript. [Line 28]
Comments 4: Line 41: (.) instead of (,) “[2,3], The global”
Response 4: We are very sorry for our careless mistake and have corrected it in the manuscript. [Line 41]
Comments 5: Line 110: It could be interesting to do a more complex statistical analysis…you could try a GLM (binomial for prevalence) to figure it out if there is some difference between places or interactions. It is not clear to me if the difference between species is because there is a source of contagion in one place and not because a single species is more prone to get a protozoan. To improve the results section (or even in methods) it could helpful to add a table with species/individuals/study location.
Response 5: Due to the social nature of most NHPs studied, collecting individual fecal samples posed challenges, our study did not collect primate related information such as sex, age, and symptoms. In addition, there were differences in the NHP species introduced in each zoo, which led to difficulties in GLM statistical analysis. We have added the occurrence of Entamoeba spp. in NHPs of 5 study zoos [Table 3, line 134], broken down the distribution of samples collected from each zoo in NHPs [Table 1, line 92] and presented their infections [Table 6, line 176] in the manuscript, and the p values have also been added to the tables. We found that the prevalence of Entamoeba spp. differed significantly among zoos, and although Entamoeba spp. differed significantly among NHP species, it did not differ significantly among NHP species in Guiyang Zoo, which may indicate that there is a source of contagion in Guiyang Zoo, due to the small sample size, this requires more research to support this point.
Comments 6: It is also a pity you didn’t count oocysts in fecal samples by Macmaster devices or similar tool as a complement, to calculate intensities that could be helpful to understand the process.
Response 6: Thank you for pointing this out. Our study picks up where the previous study left off and addresses the epidemiology of Entamoeba spp. infections in NHPs in these five zoos, detection of oocysts in fecal samples has been described in previous studies, however, due to biosecurity and other reasons, fecal samples from NHPs in some areas could not be mailed, and we converted them to fecal genomic DNA extracts for follow-up experiments, unfortunately, oocysts from Entamoeba spp. were not detected in other fecal samples. (Epidemiology and Molecular Characterization of Zoonotic Gastrointestinal Protozoal Infection in Zoo Animals in China. Animals: an open access journal from MDPI 2024, 14, doi:10.3390/ani14060853).
Comments 7: Line 178: should start with the full name if it is a dot in the final. “, E. nuttalli [11]. Entamoeba nuttalli
Response 7: Thanks for your careful checks, we have corrected this error, in addition we have examined the entire manuscript and found a similar error by changing “. E. histolytica” to “. Entamoeba histolytica”. [Line 208, 203]
Comments 8: Line 210: “(p < 0.01)” this is out of place, it’s a result.
Response 8: Agree. We have removed (p < 0.01) in the Discussion. [Line 237]
Comments 9: In Discussion or Conclusion: don’t you think is important to discuss the genetic proximity of Pan troglodytes to Homo sapiens, considering the risks of contagion of close related protozoans?
Response 9: Thank you very much for your suggestion. Due to the genetic proximity of Pan troglodytes to Homo sapiens, a discussion of the zoonotic potential of the Entamoeba spp. in Pan troglodytes is warranted, and we have added relevant discussion in the manuscript. [Line 256-261]
We sincerely appreciate the time and effort invested by the reviewers in evaluating our manuscript. We look forward to any additional feedback or suggestions. Special thanks to you for your good comments.
Yours Sincerely,
Jing Liu
Round 2
Reviewer 1 Report
Comments and Suggestions for Authors
Corrections improved the manuscript.
Author Response
Dear Editors and Reviewers,
Thank you very much for your positive response, we are pleased to hear that the revisions have improved the manuscript and addressed your previous concerns.
We appreciate the time and effort you have invested in reviewing our work and providing valuable feedback. Your expertise and insights have been instrumental in enhancing the quality and clarity of our research presentation.
Once again, thank you for your support and guidance. Please feel free to reach out if you have any further suggestions or if there's anything else we can do to refine our manuscript further.
Yours Sincerely,
Jing Liu